# Effect of Different Rotational Speeds on Graphene-Wrapped SiC Core-Shell Nanoparticles in Wet Milling Medium

**DOI:** 10.3390/ma14040944

**Published:** 2021-02-17

**Authors:** Dong Liang, Ling Yan, Kunkun Huang, Yan Li, Fangfang Ai, Hongmei Zhang, Zhengyi Jiang

**Affiliations:** 1School of Materials and Metallurgy, University of Science and Technology Liaoning, Anshan 114051, Liaoning, China; liangdong@ustl.edu.cn (D.L.); huangkunkun@ustl.edu.cn (K.H.); 2State Key Laboratory of Metal Material for Marine Equipment and Application, Anshan 114009, Liaoning, China; 2323liyan@sina.com (Y.L.); aifangfang@163.com (F.A.); 3School of Mechanical, Materials, Mechatronic and Biomedical Engineering, University of Wollongong, Wollongong, NSW 2522, Australia; jiang@uow.edu.au

**Keywords:** graphene, composites, core–shell structure, milling

## Abstract

The effects of the wet milling rotating speed on the number of graphene layers and graphene quality, and the conversion efficiency of graphite exfoliate to graphene, were investigated by scanning electron microscopy (SEM), high resolution transmission electron microscopy (HRTEM), Raman spectroscopy, Fourier transform infrared spectroscopy (FTIR), and X-ray diffraction (XRD). The results show that the number of few-layer graphene nanometer sheets (GNSs) (≤10 layers) gradually increases with the increase of rotational speed in the range of 160–240 rpm. The proportion of GNSs with 0–10 layers reaches more than 80% as the rotational speed is increased to 280 rpm. GNS defect types in the composite materials are marginal defects with minimal influence and almost no oxidation. In the range of 160–280 rpm, the intensity of graphite peak decreases and the conversion efficiency of graphene increases with the increase of rotational speed. This is the same as the experimental result obtained by HRTEM.

## 1. Introduction

Currently, graphene has great potential applications and is widely used in aerospace, electronics, energy, structural and mechanical engineering, environment, medicine, and packaging applications. The nano-graphene core-shell structure is an important research direction for the comprehensive performance and multiple functions of composite materials [1].

At present, the solid lubricant fillers used frequently mainly include graphite, carbon nanotubes, and graphene, but they have the disadvantages of low hardness and toughness. By combining solid lubricant with hard ceramic materials, the defects of low toughness and low thermal conductivity of ceramic particles can be remedied so that the lubricating effect of soft particles and the strengthening effect of hard particles can play a role together [2]. The wrapping of graphene nanosheets (GNSs) and nanomaterials can effectively prevent the polymerization of nanomaterials and broaden the application range of ceramic materials. Fadavi et al. found that graphene can promote the engulfment of silicon carbide (SiC) nanoparticles during the solidification process of A357, SiC–GNSs aluminum matrix composites and, consequently, both the toughness and strength were enhanced [3]. Zhang et al. found that sintering graphene-wrapped SiC core-shell-structured nanoparticles with aluminum-based composite materials can promote the dispersion of SiC nanoparticles. At the same time, it greatly improves the wear resistance, strength, and toughness of aluminum-based composites [4]. This also demonstrates that graphene-wrapped nanoparticles can improve many properties of metal matrix composites. At present, the methods for preparing graphene-wrapped composite materials mainly include electrostatic self-assembly method [5], aerosol phase method [6], hydrothermal synthesis [7], and emulsification method [8]. However, these methods require complex production process, reagent hazard, and high cost of production. Li et al. used silicon carbide (SiC) powder with B_4_C as a sintering aid, added 0–5.0 wt.% GNSs as experimental raw materials, and sintered at 2130 °C for 1 h without pressure to prepare GNSs/SiC composites [9]. Tang et al. studied graphene-wrapped Al–SiC-reinforced composites through finite element simulation methods. The experimental results show that the contribution of graphene flakes to thermal mismatch and Orowan strengthening mechanism is greater than the use of wrapped graphene [10]. When the particle size of the strengthening phase is smaller, the enhancement effect of the wrapped graphene is better, which is reflected in the theory of fine-grain-strengthening mechanism. Luo et al. used laser-assisted growth to prepare the core-shell SiC/GNSs nanocomposite growth process and finally formed a graphene shell (1–3 layers) around the SiC [11]. However, this method requires the use of a high-power laser to generate extreme non-equilibrium states such as ultra-high temperature (>104 K) and ultra-high pressure (>1 GPa) near the interface of the material. For the research of such a system of inherent nanostructure, Goyenola et al. used DFT to simulate the synthesis and growth of FL sulfocarbide (CS_x_) under C_m_S_n_ (m, n ≤ 2) conditions [12]. Gueorguiev et al. calculated the infrared spectrum of metcars X_8_C_12_ in the gas phase measured by the infrared resonance enhanced multiphoton ionization technique with X = V, Nb, Zr based on first principles [13]. Wang et al. [14,15] and Gai et al. [16] developed a new method for graphene exfoliation by combining ultrasound and shear in supercritical CO_2_ to prepare graphene and one-dimensional graphite nanoscrolls. This study shows that graphene can be effectively obtained by mechanically shearing and exfoliating graphite in a certain medium. Ouyang et al. used plasma-assisted dry ball milling to promote the exfoliation of graphite and reduce its disorder [17]. Gorrasi et al. widely used ball milling to prepare composites materials with nanofillers in polymetric matrices [18]. Zhang et al. developed a one-step synthesis method of graphene-wrapped nanocomposites by adding liquid as a process control agent for ball milling [19]. Compared with other methods, this method is simple, does not involve dangerous reagents, and does not need high temperature and high pressure and other extreme conditions. More importantly, the production cost is lower. The advantage of this method is that the liquid phase can be used as a process control agent to reduce the influence of temperature in the ball milling process, and the nanoparticles can be effectively dispersed to make the wrapping effect more uniform. Most importantly, the impact force between the graphite and the milling ball can be converted into shear force so that the shear force becomes the dominant force, which is more conducive to graphene peeling. Based on the experience brought by this experiment and this method, it is a problem worth exploring and discussing to change the ball milling speed and explore the influence of the rotational speed on the number of graphene layers, wrapping conditions and graphene quality [20]. At present, there is no relevant research on the preparation of graphene-wrapped SiC core-shell nanoparticles under wet milling media at different rotational speeds. This study can provide valuable experience for related research.

In this paper, graphene-wrapped SiC nanocomposites were synthesized by wet milling method using graphite and SiC as raw materials [21]. The effect of the rotational speed on the number of graphene layers, graphene quality, and conversion efficiency of graphene was analyzed.

This article is divided into three parts: introduction, materials and methods, and results. The introduction mainly collects the related research on graphene-wrapped composite materials in recent years and puts forward the research purpose of this article. The material and method part mainly explains the initial materials and specific steps of the experiment, and introduces the role of the relevant instruments in the experiment and specific parameters. The results part mainly summarizes and discusses the experimental results of SEM, HRTEM, Raman spectroscopy, FTIR, and XRD.

## 2. Materials and Methods

### 2.1. Experimental Materials

The size of graphite flakes is about 10 μm, and for SiC nanoparticles, is about 50 nm. A silicon nitride (Si_3_N_4_) milling ball was used as milling medium with a diameter of 5 mm, and the process control agent is ethanol and distilled water.

### 2.2. Experimental Method

The total weight of SiC nanoparticles and graphite flakes is 10 g, and the volume fraction of graphite flakes is 16.7%. The total mass of Si_3_N_4_ milling balls is 300 g. The mixed powder and milling balls were placed in a 500 mL Si_3_N_4_ milling jar at the same time.Ethanol (40 mL) and distilled water solution (volume ratio = 7:3) are added to the milling jar as an abrasive medium or process control agent.The milling jar is clamped in the planetary ball mill with the clamping fixture, and the machine cover is covered. The rotating speed is set at 160 rpm, and the milling time is 50 h.When the wet ball milling is finished, the powder is dried in a vacuum drying oven and then tested.The rotational speed of the ball mill is adjusted to 200, 240, and 280 rpm, and the above steps repeated for the experiment.

### 2.3. Experimental Equipment

Graphene-wrapped SiC core-shell nanoparticles were synthesized using a planetary ball mill (Retsch PM100, Verder Group, Haan, Germany) at 220 V voltage and 16 A current in one step. The movement mode of the planetary ball mill is different from that of the ordinary ball mill. Figure 1 shows the set-up of the planetary ball mill. The planetary ball mill is rotating and revolving at the same time, and the direction of movement is opposite. The whole system is composed of milling jar and its symmetrical counterweight, and rotating base [22]. In a milling process, the milling jar rotates in the direction of ①, where the rotating base driven by the rotor moves in the opposite direction ③. The milling balls run down along the jar’s wall and impact the wall as shown in the direction ②. When the other parameters are the same, changing the speed of the ball mill will cause three different motion states in the milling jar, namely, the drop type, throw type, and centrifugal type [23]. Among them, the drop type movement is mainly milling, and the impact is supplemented. The impact capacity of the throw type is larger than that of the dropping type milling medium, but the presence of process control agent can transform the impact force that is not conducive to milling into the shear force that is beneficial to milling [24]. The centrifugal type has less milling effect than the first two types of motion. The lower rotational speed can make the milling medium in a dropping or throwing mode, which is more conducive to the full milling of composite materials. 

As shown in Figure 2, in the process of wet ball milling to exfoliate the graphite layer, there are two main changes in the three-dimensional structure of graphite: one is the layer-by-layer peeling of graphite; the other is the interlayer fracture of the graphite [25]. The impact force is transformed into a shear force to make the graphite flakes exfoliate without damage with lower rotational speed. On the contrary, the impact force is large with high rotational speed, resulting in the graphite layer be crushed and refined at its fragile point, and the phenomenon of lateral fracture occurs. Zhao et al. found the shear force provided by 80 and 100 rpm was greater than that of 120 rpm using EDEM simulation software to simulate the ball milling experiments [26]. However, the time proportion of the torque received by the milling ball increases with the rotational speed.

The powder after wet ball milling is dried in a vacuum drying oven (DZF-6050, Kebeter, Beijing, China). The initial materials before ball milling and the surface morphology of the composites after ball milling were observed by high resolution scanning electron microscope (SEM, Zeiss-IGMA HD, Jena, Germany) at 5 kV voltage. The dispersion of graphene-wrapped SiC core-shell nanoparticles and the number of graphene layers at different rotational speeds were observed by high resolution transmission electron microscope (HRTEM, JM-2100, NEC, Tokyo, Japan) at 200 kV. For the detection of the conversion efficiency of graphite exfoliation to graphene, the X’Pert Powder X-ray diffractometer (XRD, X’Pert Powder, PANalytical, Beijing, China) was used to conduct qualitative and quantitative analysis of the composite material within the range of 2θ from 10–80°. The defect types and sizes of carbon-related materials were evaluated by Raman spectrometer (Xplora Plus, HORIBA, Shanghai, China) in the spectral range of 1000–2000 cm^−1^. The chemical bonds and functional groups of the composites were detected using a Fourier transform infrared spectrometer (FTIR, Nicolet iS10, THERMO, Massachusetts, MS, USA) in the spectrum range of 750–4000 cm^−1^, which provided a favorable basis for the quality detection of graphene.

## 3. Results

### 3.1. Surface Morphology of Composite Material before and after Ball Milling

Figure 3 shows the SEM morphology of graphite flakes and SiC nanoparticles of the initial material. Figure 3a shows graphite flakes, most of which have particle sizes about 10 μm. The SiC nanoparticles in Figure 3b have spontaneous agglomeration. Due to the large specific surface area of SiC nanoparticles, excessive surface energy is generated, making it tend to change to a low energy surface area state under the action of surface tension.

Figure 4 shows the SEM morphology of graphene-wrapped SiC nanoparticles randomly selected after wet ball milling at different speeds. By observing the composites milled at different speeds, it is found that the graphite flakes are obviously thinner and exfoliated into GNSs layer by layer, which are light gray translucent and evenly dispersed in SiC nanoparticles. As the rotation speed increases, GNSs become smaller and thinner. Among them, the GNSs in Figure 4d are thinner and smaller than that of the low speed.

### 3.2. Dispersion of GNSs and SiC in the Liquid Phase

The HRTEM morphologies of graphene-wrapped SiC nanoparticles at 160 and 280 rpm were observed (magnified about 500,000 times) as shown in Figure 5. It can be seen from this figure that the graphene layer is uniformly dispersed around the SiC nanoparticles, so the composite material has great dispersibility at different rotational speeds. This is mainly due to the fact that the dispersibility of SiC nanoparticles in 70% alcohol and 30% distilled water solution is superior to that of other mixed solutions [19]. Therefore, GNSs and SiC nanoparticles fully contact under the depolymerization effect of wet ball milling, which promotes the formation of graphene-wrapped SiC core-shell structure.

### 3.3. The Influence of Different Rotational Speeds on the Number of Graphene Layers

#### 3.3.1. The exfoliation of GNSs in the Presence of SiC Nanoparticles

As shown in Figure 6, the HRTEM (approximately 10^6^ times magnification) morphology of graphene-wrapped SiC nanoparticles at different speeds were randomly selected. From this figure, the graphene-wrapped SiC core-shell nanoparticle microstructure can be seen clearly. Single-layer graphene thickness is 0.335 nm and the number of layers of multilayer graphene can be calculated by HRTEM. It can be seen from Figure 6a that most of the few-layer graphenes have 5–10 layers, and those with more than 10 layers of GNSs account for a large proportion. In Figure 6b, the few-layer graphene is mainly composed of 0–5 layers, and those with thick GNSs are significantly reduced. In Figure 6c, the GNSs of 0–5 layers occupy an absolute proportion, and there will still be a small amount of thick GNSs. In Figure 6d, almost all GNSs are less than 5 layers, and the probability of core-shell structure is the highest.

Rafael Tadmor deduced the mathematical expression of the interaction energy between macroscopic objects with common geometric shapes by establishing the London–van der Waals interaction energy among different geometries [27], which provided a basis for the energy transfer of graphite flakes by ball milling. Figure 7 shows the energy model of the London–van der Waals model between two parallel slices.

This model can be compared to the graphite flakes and single-layer graphene in this experiment. When the shear force between the graphite flakes given by the milling ball is greater than the van der Waals force between the graphite flakes, the graphene will exfoliate from the graphite flakes. The calculation formula is as follows [27]:(1)$$E_{A}=-\frac{A}{12\pi }\left[\frac{1}{d^{2}}+\frac{1}{{\left(d+h_{1}+h_{2}\right)}^{2}}-\frac{1}{{\left(d+h_{1}\right)}^{2}}-\frac{1}{{\left(d+h_{2}\right)}^{2}}\right]$$
where *A* is the Hamaker constant (2.38 × 10^−19^ j).

Most of the graphite flakes used in this experiment are 10 μm, which was set as the thickness of *h*_1_, *h*_2_ is the thickness of single-layer graphene at 0.335 nm, and *d* is the distance between two graphite layers at 0.335 nm. The calculation shows that the energy required to exfoliate a single layer of graphene from the graphite flake is 4.2 × 10^−13^ j. According to this method, the energy required for exfoliate 10 layers (3.35 nm) and 20 layers (6.7 nm) can be calculated.

The energy transfer from the milling ball to graphene was evaluated with the increase of ball milling speed. The calculation formula is as follows [28]:(2)$$△E=\frac{\pi }{12}\cdot \rho_{mm}\cdot d_{mm}{}^{3}\cdot \left(v^{2}{}_{mm1}-v^{2}{}_{mm2}\right)\cdot \left(1+\frac{Y_{P}}{Y_{MM}}\right)$$
where: *ρ_mm_*—The density of Si_3_N_4_ milling ball is 3100 kg/m^3^;

*d_mm_*—The diameter of the milling ball is 5 mm;

*v*_mm1_—The tangential velocity v_mm1_ of the faster milling media is 0.85 m/s;

*v_mm2_*—The tangential velocity v_mm1_ of the slower milling media is 0.425 m/s;

*Y_P_*—The Young’s modulus of the final product is 230 GPa; 

*Y_mm_*—The Young’s modulus of the milling media is 388 GPa.

The energy provided by the milling medium is 9.67 × 10^−8^ j, which is enough to exfoliate all sizes of graphite into graphene. While ensuring that the structure of the graphite flake is not destroyed, as the speed of the ball milling increases, the tangential speed of the milling media is also constantly increasing. Therefore, the energy transferred from the milling media to the graphene is also increasing. When the ball milling speed is faster, the graphene will be exfoliated into fewer layers.

#### 3.3.2. Distribution of Graphene Layers at Different Speeds

The thickness of the produced GNSs was estimated from the number of layers of the GNSs in their TEM images. The layer number distribution of GNSs was obtained from 40 random HRTEM images, either standing alone or wrapping nanoparticle. As shown in Figure 8a, when the rotational speed is 160 rpm, about 60% of GNSs have 10–20 layers, and the proportion of GNSs with 0–10 layers is relatively small. At this speed, the impact provided by the milling balls and SiC nanoparticles is small, so the shear force converted by the liquid phase is not enough to exfoliate more graphite flakes into GNSs with fewer layers. As shown in Figure 8b, when the rotational speed is 200 rpm, the proportion of GNSs with 10–20 layers is reduced. Among them, compared with the former, the proportion of GNSs with 10–15 layers is significantly reduced, while the proportion of GNSs with 5–10 layers is increased from 29% to 41%. As shown in Figure 8c, when the rotational speed is 240 rpm, the distribution of the number of layers is basically similar to that of 180 rpm. GNSs of 0–10 layers are significantly increased, and GNSs of 5–10 layers account for almost half of the total. When the rotational speed reaches 280 rpm, as shown in Figure 8d, the GNSs of 0–10 layers account for more than 80%, of which the GNSs of 0–5 layers account for the majority. This is mainly due to the increased rotational speed of the milling ball and SiC nanoparticles providing more energy.

#### 3.3.3. Formation of Core-Shell Structure

In the process of counting the number of graphene layers, it was found that the number of GNSs that can be wrapped on SiC nanoparticles is often 7 layers or less, and the thick GNSs are almost all scattered in the composite material. Shown in Figure 9 is the schematic diagram of wrapping method of graphene on SiC nanoparticles. Through theoretical analysis, it can be seen that the GNSs with fewer layers (≤7 layers) are thin and tend to bend and roll [29], which directly promotes the in situ wrapping of graphene on SiC nanoparticles. GNSs with more layers are thicker and more rigid and not easy to bend [30]. At this time, the wrapping method can only be superimposed by layers of GNSs to wrap SiC nanoparticles around GNSs. In addition, most of the SiC nanoparticles are separately wrapped, which is mainly due to the good dispersibility of GNSs and SiC nanoparticles in ethanol–water solution [19].

The performance of graphene in tribological behavior is not only affected by the number of layers, but also related to the types of defects in grapheme. Wang et al. used molecular dynamics method to study the effect of vacancy defect and SW defect on the friction force of graphene. Simulation results show that the friction on graphene with both kinds of defects were obviously larger than perfect graphene in the defect regions, which was attributed to the increasing energy dissipation resulted from the extended interface barrier potential of graphene due to vacancy [31]. Therefore, it is of great significance to study the defects and defect types of graphene [32,33].

### 3.4. The Influence of Different Rotational Speeds on the Quality of Graphene

Shown in Figure 10a is the Raman spectrum of the initial material graphite flakes. Since pure graphite has almost no defects, no obvious D peak appears. Figure 10b shows the Raman spectra of four composite materials at different rotational speeds. The three peaks are D peak (1350 cm^−1^), G peak (1580 cm^−1^), and D’ peak (1620 cm^−1^) [34]. 

The vibration direction of the atom parallel to the plane of graphene is expressed as (i), the vibration direction perpendicular to the plane of graphene is expressed as (O), the vibration direction parallel to the direction of A–B carbon bond is expressed as longitudinal (L), and the vibration direction perpendicular to the direction of A–B carbon bond is expressed as (T). The in-plane vibration of sp^2^ carbon atom and its interaction with iTO and iLO optical phonons promote the generation of G peak, which is a very rare Raman scattering process in graphene. As the D peak and G’ peak of the second-order double resonance Raman scattering process, iTO optical phonons will be generated under two inelastic scattering between valleys. Among them, the D peak is related to the scattering between iTO phonons and defect valleys, and the number is one. According to experience, the appearance of the D peak is caused by the deviation of the lattice of the graphite flake layer from the Brillouin zone, so the D peak reflects the defects of the graphene lattice. The G peak is due to the in-plane vibration produced by the σ bond bonding of carbon atoms [35]. 

The intensity ratio I_D_/I_G_ is a recognized index for evaluating the strength of defects in carbon-related materials [36,37]. As can be seen from Figure 10b, the I_D_/I_G_ intensity ratio of initial graphite is about 0.03, and the I_D_/I_G_ intensity ratios in the experiments at 160, 200, 240, and 280 rpm are 0.41, 0.57, 0.59, and 0.68, respectively. Its value is much lower than that of the graphene prepared by redox method (I_D_/I_G_ > 1) [38] and dry ball milling (I_D_/I_G_ > 1.5) [17]. From the experimental results of HRTEM, it can be seen that with the increase of the rotational speed, the proportion of the few-layer graphene continues to increase. Therefore, the increase of I_D_/I_G_ strength is related to the gradual thinning and exfoliation of graphite flakes into graphene and the generation of defects. There is no obvious D’ peak in the experiment with the rotational speed of 160 rpm. When the rotation speed is increased to 200 rpm, a faint D’ peak appears, which is a small part of the G peak and gradually increases with the increase of the rotational speed. According to the ratio of I_D_/I_D’_, information about the nature of the defect can be obtained. The defect types can be divided into sp^3^ hybridization-related defects (≤13), vacancy defects (≤7), and edge defects (≤3.5) [39]. According to the calculation, the defect types are all edge defects that have the least impact on the quality of graphene.

Figure 11 shows the initial material SiC nanoparticles and graphite flakes, and graphene-wrapped SiC nanoparticles at different milling speeds measured in FTIR. There is a weak C–O peak near 1086 cm^−1^, and there is no C=O, –OH stretching vibration peak, and C–O–C vibration absorption peak related to graphene oxide [40,41]. Two strong peaks appearing near the wave numbers of 2335 and 2365 cm^−1^ are the absorption peaks of CO_2_ in the air.

### 3.5. The Influence of Different Rotational Speeds on the Conversion Efficiency of Graphite Exfoliate to Graphene

Figure 12 shows the XRD test results of initial materials and graphene-wrapped SiC nanoparticles at different ball milling speeds. The curve has a higher intensity diffraction peak at 2θ = 26.5°. According to the Bragg formula (2dsinθ = nλ), the distance between crystal planes at (002) is about 0.34 nm, which indicates that this is the graphite peak. With the increase in ball milling speed, the intensity of graphite peak gradually decreases and the conversion rate of graphene gradually increases. This usually indicates that the size of the thick graphite flakes is continuously reduced and exfoliated into thin GNSs, and it is also consistent with the experimental results observed by HRTEM. With the increases of rotational speed, the graphite peaks gradually broaden. This is usually related to the thinning of the graphite flakes and the resulting defects [42]. The other peaks are all SiC nanoparticles. When the rotational speed is 160–240 rpm, the peak value does not change significantly. When the rotational speed is increased to 280 rpm, the peak intensity of SiC is significantly reduced. This is due to the high rotational speed and the SiC nanoparticles being ground into granules.

It can be seen that the graphene conversion efficiency of the composites is very high after 50 h milling. With the increase of milling speed, the conversion efficiency of graphene increases gradually. At the same time, it can ensure the integrity of SiC nanoparticles to a certain extent.

The conclusions obtained using SEM, HRTEM, Raman spectroscopy, FTIR, XRD, and other characterization methods for graphene-wrapped SiC core-shell nanomaterials may be equally applicable to other similar graphene-wrapped inherently nanostructure systems.

## 4. Conclusions

In this paper, the rational speed of the ball milling was changed by wet ball milling, and the influence of different rational speeds on the number of graphene layers, quality, and conversion efficiency was analyzed.

The results show that the graphene is well dispersed in SiC nanoparticles when the ball milling speed is in the range of 160–280 rpm, and graphene-wrapped SiC core-shell nanocomposites were synthesized.With the increase in ball milling speed, the proportion of few-layer graphene (≤10 layers) increases. When the rotational speed is 160–240 rpm, the proportion of few-layer graphene increases from 45% to about 70%. When the rotational speed is increased to 280 rpm, the proportion of few-layer graphene increases to more than 80%. At the same time, it is found that most of the GNSs wrapped on SiC nanoparticles are ≤7 layers.When the rotational speed is 160–280 rpm, the obtained graphene has relatively small defects, and the defect type is the edge defect that has the least impact on the quality of the graphene and, at the same time, it is almost not oxidized. Therefore, the graphene produced by wet ball milling has better quality.With the rotational speed increases, the peak intensity of graphite gradually decreases and the conversion efficiency of graphene gradually increases. When the rotational speed reaches 280 rpm, the conversion efficiency of graphene has reached an ideal state.The graphene-wrapped SiC core-shell nanocomposites prepared by wet ball milling have high quality. When there are more graphene layers, the anti-friction ability and self-lubricity are stronger. Therefore, in practical application, when the main purpose of the required materials is to reduce friction and wear, a lower ball milling rotational speed can be selected. If the composite material is for the purpose of improving the toughness, thermal conductivity, and optical properties, the ball milling rotational speed can be appropriately increased.

## Figures and Tables

**Figure 1 materials-14-00944-f001:**
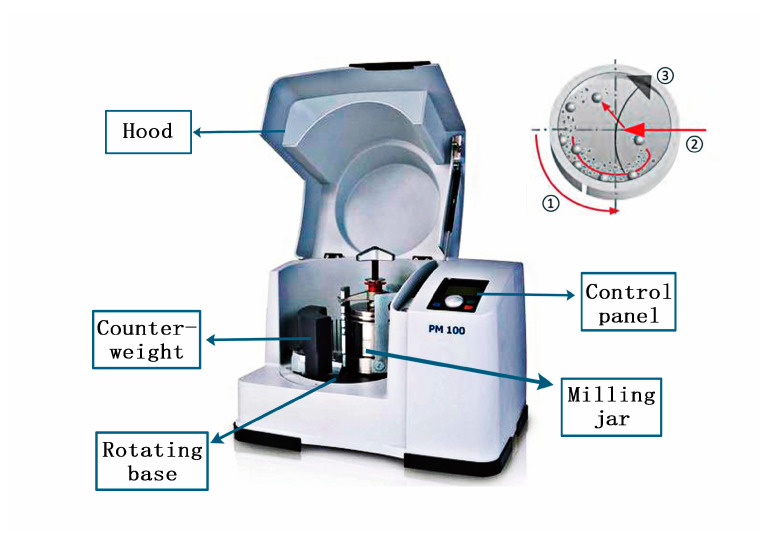
The set-up of the planetary ball mill.

**Figure 2 materials-14-00944-f002:**
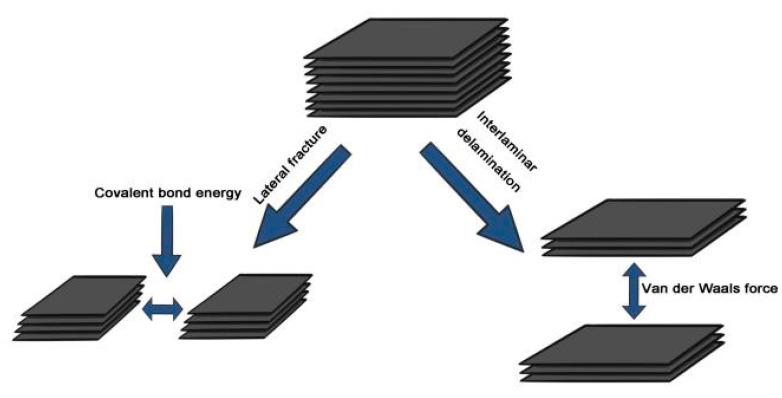
Structure change of graphite during wet ball milling.

**Figure 3 materials-14-00944-f003:**
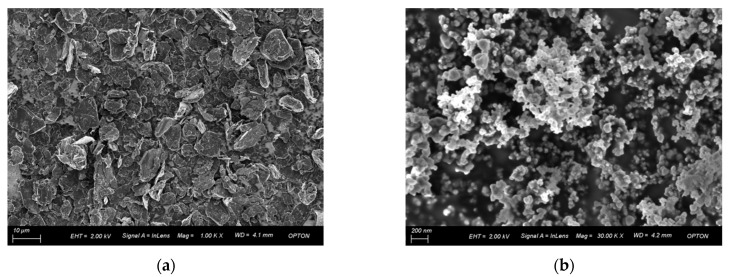
SEM morphology of graphite sheet and SiC nanoparticles. (**a**) Graphite sheet; (**b**) SiC nanoparticles.

**Figure 4 materials-14-00944-f004:**
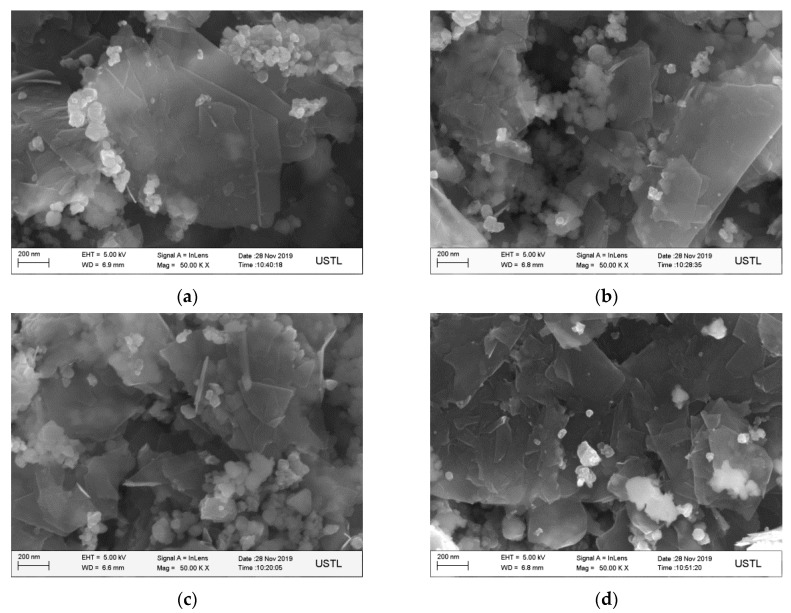
SEM morphology of graphene-coated SiC nanoparticles at different rotational speeds. (**a**) 160 rpm; (**b**) 180 rpm; (**c**) 240 rpm; (**d**) 280 rpm.

**Figure 5 materials-14-00944-f005:**
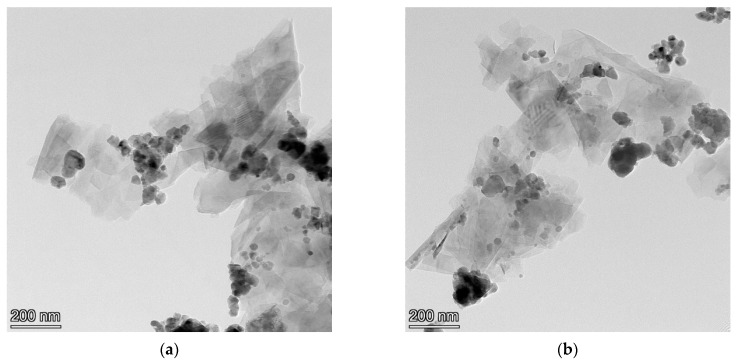
HRTEM topography of four different speeds at (**a**) 160 rpm; (**b**) 280 rpm.

**Figure 6 materials-14-00944-f006:**
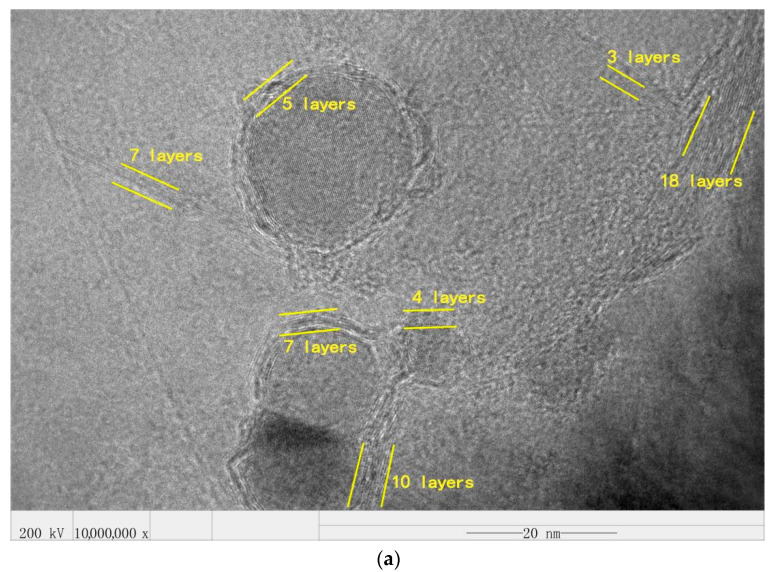
Structure of graphene-wrapped SiC nanoparticles randomly selected at different rotational speeds in HRTEM. (**a**) 160 rpm; (**b**) 200 rpm; (**c**) 240 rpm; (**d**) 280 rpm.

**Figure 7 materials-14-00944-f007:**
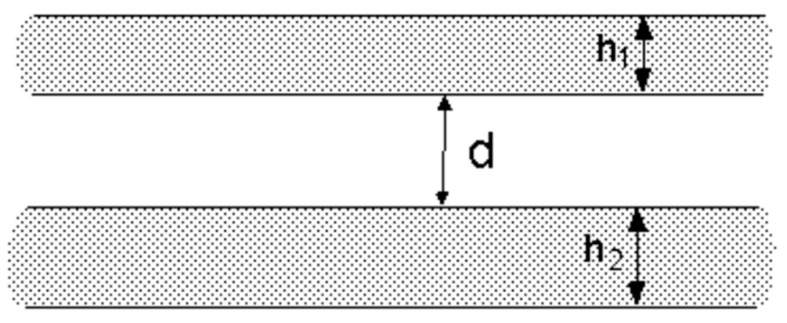
Energy model of London–van der Waals model between two parallel plates.

**Figure 8 materials-14-00944-f008:**
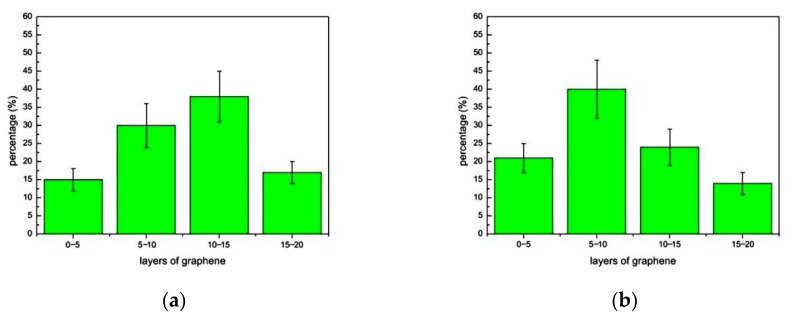
Histogram of graphene layers distribution in composite under different rotational speeds. (**a**) 160 rpm; (**b**) 200 rpm; (**c**) 240 rpm; (**d**) 280 rpm.

**Figure 9 materials-14-00944-f009:**
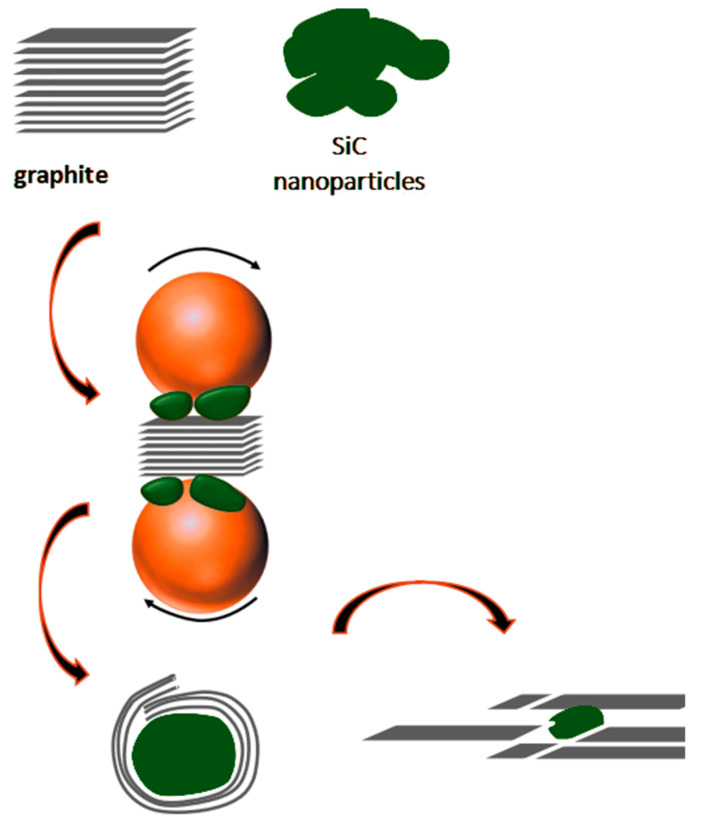
Schematic diagram of wrapping method of graphene on SiC nanoparticles.

**Figure 10 materials-14-00944-f010:**
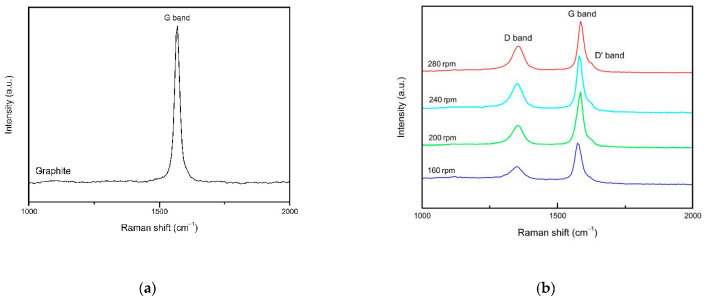
Raman spectra of graphite and composite materials at different speeds: (**a**) Raman spectrum of the initial material graphite flakes; (**b**) Raman spectra of four composite materials at different rotational speeds.

**Figure 11 materials-14-00944-f011:**
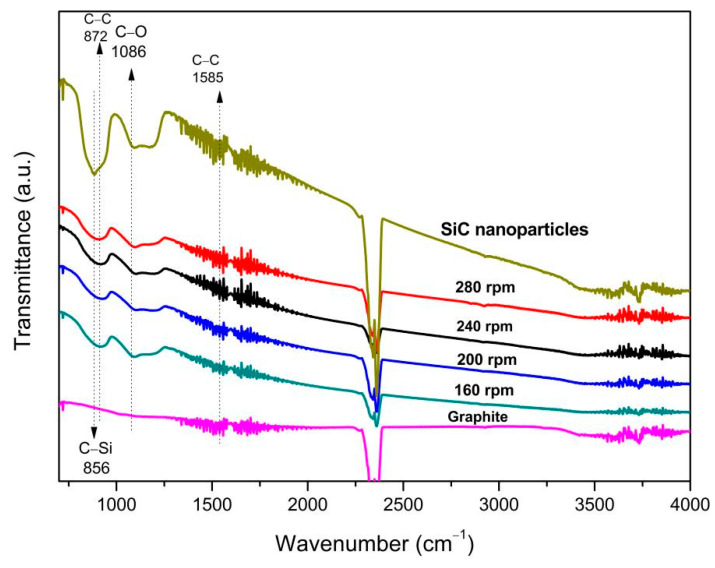
FTIR spectra of the composites at different rotational speeds and initial materials.

**Figure 12 materials-14-00944-f012:**
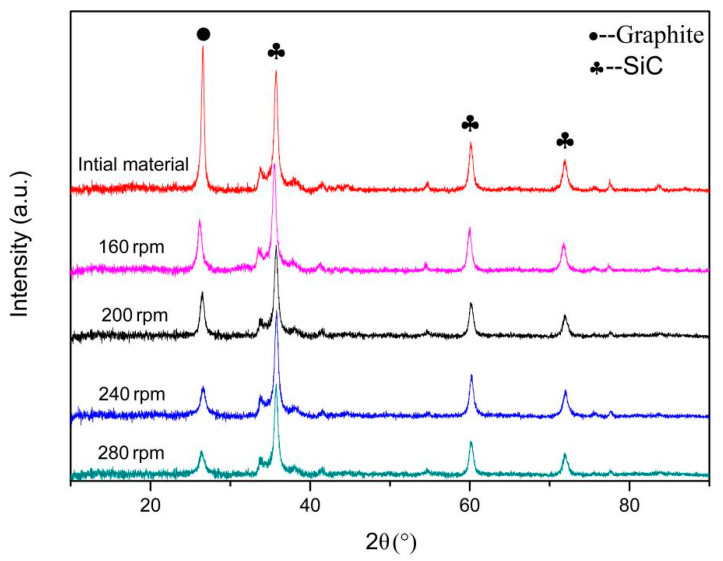
XRD patterns of the composites at different rotational speeds and initial materials.

## Data Availability

Data sharing not applicable.

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
