# Peer review of "Effect of Different Rotational Speeds on Graphene-Wrapped SiC Core-Shell Nanoparticles in Wet Milling Medium"

_materials, 2021, doi:10.3390/ma14040944_

Round 1
Reviewer 1 Report
The manuscript received for evaluation, titled 'Effect of different rotational speeds...' by Z. jiang et al. describes what happening on milling at different speed in the exfoliation process of graphene, mainly how this affects the number of layers and therefore the quality of graphene, making it more suitable for different applications.
The work is performed in a good manner, starting with the morphology of composite material before and after ball milling. Techniques used for following the processes are SEM, TEM, IR, XRD, Raman. The experimental equipements are described in details. The manuscript can be of high interest for industry, as it give some details about a procedure that can be applied for gaining high yields high quality graphene materials. The manuscript has 31 references that cover mainly the area of the work. English should be checked carefully, for example in Figure 10b- please correct 'Ranman'.
As a suggestion, maybe it was better to have also some elemental or surface analysis of the graphite/graphene before and after milling, in order to see if it can be observed an increase in the oxygen content of the sample, because during process it is possible to be formed some oxygen-rich groups. However, this doesn't affect the main work.
In conclusion, the manuscript can be accepted for publication as it is.
Reviewer 2 Report
In this paper, the authors synthesize graphene wrapped SiC nanocomposites by wet milling method using graphite and SiC as raw materials. They study the effect of the rotational speed on the number of graphene layers, graphene quality and conversion efficiency of graphene. They show that the number of few-layer graphene nanometer sheets (≤10 layers) increases with the increase of the wet milling rotational speed and that the milling process produces mainly edge defects.
The paper is clear and contains useful information. It can be suitable for publications in the journal after a revision.
The authors should consider the following points:
- In the introduction, the authors should better motivate their work. They should emphasize the importance of fabricating graphene wrapped SiC core-shell structures and their possible uses.
- “At present, the methods for preparing graphene-wrapped composite materials mainly include electrostatic self-assembly method, aerosol phase method, hydrothermal synthesis, and emulsification method.” I suggest supporting this sentence with appropriate references.
- “Ouyang et al. used plasma-assisted dry ball milling to promote the exfoliation of graphite and reduce its disorder [8].” The authors could mention also that ball milling has been extensively used to prepare composites materials with nanofillers in polymetric matrices (see for instance https://doi.org/10.1016/j.compositesb.2017.10.020)
- “The performance of graphene in tribological behavior is not only affected by the number of layers, but also related to the types of defects in graphene.” Be more specific here. How defects affect the performance of graphene in tribological applications?
Reviewer 3 Report
This manuscript is dedicated to a study of the effect of different rotational speeds on graphene wrapped SiC core-shell nanoparticles in wet milling medium. The proposed manuscript contributes an original work on this prototypical material system, succeeding in a well-conceived and adequate characterization effort. In addition, the authors provide a good motivation behind such a research line which is backed in this manuscript by careful analysis of variety of structural and mechanical issues arising in this system. The analysis provided is with high degree of clarity. The presentation of the results is concise and convincing, easy to read/perceive and all characterization techniques that are used are adequately described and critically discussed.
Such work is really timely, while especially useful to the broader community because of the didactic way the data was discussed and interpreted in the right context of graphene wrapped systems based on SiC, carbon etc., there are other examples in the literature but by far insufficient yet. In my opinion, the manuscript is bound to quickly attracting significant research interest also in relation to nanoparticles/core-shell systems of similar nature.
There are excellent figures too.
There are some very minor aspects/questions to this, otherwise excellent manuscript, that need attention; thus, it is acceptable for publication after a minor revision:
1: Introduction and the methods sections are very well written. However, for the broadness of understanding of these and similar INHERENTLY nanostructured systems it should be mentioned that modelling/DFT of carbon-containing and also Si containing systems is available for to studying bonding/van der Waals interactions etc. Such works as [J. Phys. Chem. C2012, 116, 21124−21131 and PHYSICAL REVIEW B68, 241401 (2003)] should not be missed.
2: On page 10 the features of Raman spectra are discussed also in relation to bonding changes occurring in the system. However, more analysis of what implications on the C bonding have the observer peak characteristics should be provided in the text. Currently, concerning this issue, the text is just descriptive
3: Generally speaking, it should be emphasized that the results and the discussion presented in this work may be applicable to other similar graphene-wrapped inherently nanostructured systems.
4: Conclusions are nicely written but they may be improved if they invoke an explicit reference to the possible applications. Currently, applications are only mentioned in the abstract/intro. Thus, the final message of this manuscript will be even better transmitted. This will also make present work better citable.
5: Spell-check and stylistic revision of the English of the paper is still necessary.
Reviewer 4 Report
The reviewer comments of the paper «Effect of different rotational speeds on graphene wrapped SiC core-shell nanoparticles in wet milling medium»
- Reviewer
The authors presented an article «Effect of different rotational speeds on graphene wrapped SiC core-shell nanoparticles in wet milling medium». However, there are several points in the article that require further explanation.
Comment 1:
- Introduction
The introduction needs to be rewritten.
Figures 1 and 2 are best placed in section 2 as a description of the method under study. And in the introduction, it is better to focus on a review of previously published works on the method under study.
Show more articles especially from the last 5 years. Try to reinforce the relevance of your research. It is important to more clearly define why they chose graphene wrapped SiC for research.
It is necessary to more clearly define the "white spots". That has not been previously investigated by other scientists.
However, at the end of the article, list briefly what has been done in each section of the article.
Comment 2:
- Materials and Methods
Show on the figure a general view of the experimental setup.
Comment 3:
- Result
Quality and resolution for all figures need to be improved.
Are all the formulas in the article original? If not needed appropriate citations.
Comment 4:
It will be useful to add a section of Nomenclature in which to sign all the physical quantities and abbreviations encountered in the article. There are many physical quantities in the text and such a section will help to find the description of the necessary element.
For example,
dmm : The diameter of the milling ball (mm)
SEM : Scanning electron microscopy
etc.
Comment 5:
Conclusions.
In addition, it is necessary to more clearly show the novelty of the article and the advantages of the proposed method. What is the difference from previous work in this area? Show practical relevance. Conclusions should reflect the purpose of the article.
The article is interesting. Authors should carefully study the comments and make improvements to the article step by step. After major changes can an article be considered for publication in the "Materials".
Round 2
Reviewer 4 Report
Figure 1 needs to be redrawn. Now it looks like it was scanned from an old directory. Add details to figure 1. It is useful to draw it in 3D and color. Show the main vectors of speeds and feeds. Where is the blank? Where is the cutting tool. What kind of milling scheme was used, etc.
Describe in detail how it all works.
After that, the article can be published.
